# *Cetobacterium* Is a Major Component of the Microbiome of Giant Amazonian Fish (*Arapaima gigas*) in Ecuador

**DOI:** 10.3390/ani8110189

**Published:** 2018-10-24

**Authors:** Carolina Ramírez, Jaime Coronado, Arturo Silva, Jaime Romero

**Affiliations:** 1Laboratorio de Biotecnología, Unidad de Alimentos, Instituto de Nutrición y Tecnología de los Alimentos (INTA), Universidad de Chile, Avda. El Líbano 5524, Santiago, RM, Chile; carolina.ramirez.saavedra@gmail.com (C.R.); jaime.coronados@gmail.com (J.C.); 2Acuatilsa, 9 de octubre, El Puyo, Pastaza 160150, Ecuador; silvaartu@hotmail.com

**Keywords:** microbiome, high-throughput sequencing, *Arapaima gigas*, *Cetobacterium*

## Abstract

**Simple Summary:**

*Arapaima gigas* is a large, air-breathing, giant fish found in Amazonian rivers, a characteristic that gives this species an advantage in oxygen-deprived waters. It has a very attractive potential for aquaculture in the Amazon region due to its many advantages, including a fast growth rate that approaches 10–15 kg/year. Here, we describe the gut microbiome of *Arapaima* to understand the potential contribution of this bacterial community to the growth of this fish.

**Abstract:**

*Arapaima gigas* is a large air-breathing fish found in Amazonian rivers, a characteristic that gives this species an advantage in oxygen-deprived waters. It shows high potential for aquaculture in the Amazon region due to its fast growth rate that approaches 10–15 kg/year. The aim of this study was to explore the composition of the intestinal bacterial community of *Arapaima gigas* reared in Ecuador using 16S rRNA gene high-throughput sequencing. The analysis revealed significant differences in alpha diversity indices (*p* < 0.05) and differential distribution of minor components of the intestinal microbiome between small and large fish. However, components with greater relative abundance, such as *Cetobacterium,* are found in similar proportions.

## 1. Introduction

*Arapaima gigas* (Cuvier, 1829), also called paiche or piracucu, is a giant Arapaimidae native to the Amazon River basin, which can reach up to 3 m in length and 250 kg in weight and is found in Brazil, Ecuador, Bolivia, Colombia, and Peru. This species has excellent attributes for aquaculture, including a high growth rate of nearly 10–15 kg per year [1]. Information from the Food and Agriculture Organization of the United Nations (FAO) on the production of this species have only been reported by Brazil and Peru—in 2015, production totalled 8522 tons [2]. The intestinal microbiota of vertebrates is composed of a diverse population of microorganisms that present a broad range of metabolic activities, and may contribute to various metabolic processes of the host. Thus, microbiota may contribute enzymes to complement digestive processes and vitamins to enhance nutrition. They may also play a role in the defence against pathogenic microorganisms by preventing colonization, competing for nutrients and adhesion sites, producing antimicrobial substances, and may also modulate the host immune system [3]. In this context, understanding the composition of the microbiota could provide relevant information regarding the management of the species’ food requirements in the development of a sustainable aquaculture. It could also be useful for manipulating microbiota during different growth stages in aquaculture systems to prevent pathogenic infection or to improve nutrition. Although microbiota modulation has been studied by probiotic supplementation, the use of autochthonous probiotics seems to be more beneficial than the use of probiotics from other sources of isolation [4]. The aim of this study was to explore the composition of the bacterial community of *Arapaima gigas* reared in Ecuador using high-throughput sequencing.

## 2. Material and Methods

### 2.1. Sample Collection

A total of 7 fish were used in this study. These fish were collected from Acuatilsa farm, which is located in Puyo-Pastaza, Ecuador. The intestinal contents were collected from small (*n* = 3, 70–95 g body weight) and large (*n* = 4, 3–7 kg body weight) *Arapaima gigas* individuals. Using aseptic techniques, we separated the digestive tract from the abdominal cavity, and compressed the hindguts to express the intestinal contents into a sterile container. These samples were frozen at −20 °C until analysis. 

### 2.2. DNA Extraction

Nucleic acid extraction was conducted using an MO BIO PowerFaecal^®^ DNA Isolation Kit (MoBio Laboratories Inc., Carlsbad, CA, USA), following the manufacturer’s instructions, with an initial enzymatic treatment using lysozyme and proteinase as described in [5]. DNA concentration was determined using the Qubit^®^ dsDNA HS Assay kit (Life Technologies, Grand Island, NY, USA). The V2–V4 region of the 16S rRNA gene was amplified by the fusion primer method using the primers 341F and 788R. The amplicons we obtained were purified with the QIAquick^®^ PCR Purification kit (Qiagen, Valencia, CA, USA). DNA sequencing was performed via an Ion Torrent (Life Technologies, California, CA, USA).

### 2.3. Bioinformatics and Statistical Analysis

Sequencing reads of the 16SrRNA gene were processed and analyzed using UPARSE [6]. The quality of the reads was assessed using FastQC software, and the reads were filtered by quality and length by using the USEARCH algorithm. After the resulting quality-filtered FASTA files were merged, the sequences were trimmed to 160 bp, sorted by abundance, and singletons were discarded. The reads were then clustered into operational taxonomic units (OTUs) based on 97% identity using UPARSE with the “cluster_otus” command in USEARCH. OTUs containing <5 sequences were also removed from the dataset. Taxonomy was assigned via the SINTAX approach implemented in USEARCH. Taxonomy assignment of the OTUs was performed by comparing sequences, with the RDP training set v16 [7] as the reference database. The R package Phyloseq [8] was used for data analysis and plotting. Differential abundance of bacterial components between small and large *A. gigas* was assessed using the Wald test, and *P* values were corrected using the Benjamini–Hochberg false discovery rate method using the R package DESeq2 [9]. Microbiota data have been deposited in the Sequence Read Archive of the National Center for Biotechnology Information (Sequence Read Archive (SRA), NCBI, Bethesda, MD, USA) under the accession ID SRP152854.

### 2.4. Ethical Notes

The study was conducted in accordance with the guidelines of the Bioethics and Biosecurity Committee of the Instituto de Nutrición and Tecnología de los Alimentos (INTA) at Universidad de Chile. The ethical approval code is FCYT2/17JR.

## 3. Results

In total, 193,139 raw reads were obtained after sequencing. After the initial quality filtering process, 56,430 sequences were retained. The mean read depth per sample was 8062 ± 1795 sequences per sample. In all samples, a total of 168 OTUs were detected. Within these OTUs, all those with fewer than 5 sequences, those assigned as *Cyanobacteria*, and sequences that were unclassified at the kingdom level were eliminated—thus, 84 OTUs were retained. Table 1 shows the calculated alpha diversity of the bacterial composition within each sample as expressed through the observed OTUs, Chao1, Shannon, and Simpson indices. The normal distribution of each index value was determined using the Shapiro-Wilk test, and subsequently, the t-test was applied for comparison. This analysis showed that small fish have alpha diversity indices that are significantly higher than those of large fish.

Composition of the intestinal microbiota of both small and large *A. gigas* is dominated by two phyla—*Fusobacteria* and *Firmicutes* (Figure 1A), which together represent approximately 100% of the relative abundance of each sample. Furthermore, the prevalent classes in both cases correspond to *Fusobacteriia* and *Clostridia*. Composition of microbiota at the genus level is represented by two genera, *Cetobacterium* and *Romboutsia* (Figure 1B). Relative abundance of *Cetobacterium* ranged from 55 to 82% in small fish and from 58 to 87% in large fish. This genus was followed by *Romboutsia* with 2.3–13% and 1.3–36%, respectively. Other genera observed in minor abundances were *Clostridium sensu stricto*, *Hespellia*, *Sphingobacterium*, and *Staphylococcus*. Interestingly, the significant differences in composition of the microbiota between small and large fish were observed in these minor components (Figure 2).

## 4. Discussion

Our results clearly indicated that the intestinal microbiota of *A. gigas* is dominated by the genus *Cetobacterium*. This genus has been identified as a component of the microbiota of other freshwater fish, such as *Oreochromis niloticus* [10] and *Cyprinus carpio* [11]. Interestingly, *Cetobacterium* has been observed in high relative abundance (72–94%) in *Lepomis macrochirus*, *Micropterus salmoides*, and *Ictalurus punctatus* [12]. It has been reported that *Cetobacterium* isolated from the intestine of freshwater fish produces vitamin B-12 [10]. The importance of this genus in the production of vitamin B-12 in fish was revealed in a study of microbiota of the Japanese eel, ayu, carp, tilapia, goldfish, and catfish—specifically, carp and tilapia did not require dietary vitamin B-12, and they showed high levels of *Cetobacterium* (namely, *Bacteroides* type A) and vitamin B-12 in their intestinal content [13]. A recent study by Hao et al. [14] described that the abundance of *Cetobacterium* was negatively correlated with *Bacteroides,* and this balance might be modulated by their diet. In fact, *Cetobacterium* decreased drastically due to dietary changes in grass carp *Ctenopharyngodon idella.* According to our results, abundance of *Bacteroides* is low with respect to *Cetobacterium.* Therefore, studies about microbiota and diet are necessary to understand the possible beneficial presence of this vitamin-producing bacterium. On the other hand, the genus *Romboutsia* corresponds to a recently created taxon [15] that includes some strains that were previously classified as *Clostridium*. Genomic information revealed that the sequenced strain of this taxon showed a limited capacity to synthesize amino acids, vitamins, and pathways for the utilization of simple carbohydrates. Considering this data, the role of this bacterium in the gut of *A. gigas* could be less relevant in terms of the supply of vitamins and cofactors.

The significant bacterial genera presented in small fish were *Bacteroides*, *Sphingobacterium*, *Corynebacterium*, *Staphylococcus*, and *Hespellia*. Carbohydrate fermentation occurs by members of the genus *Bacteroides*, which are known to be short-chain fatty acid producers with an important role against gut inflammation [16]. On the other hand, members of the genus *Corynebacterium* isolated from fish have been highlighted regarding the production of amino acids [17].

Previous studies have described the changes in dominant populations of fish microbiota during growth [18]. In the case of coho salmon, studies suggest that stable gastrointestinal microbiota could be acquired in the juvenile stage, where microbiota showed a similar composition in several sizes examined [19]. Our results coincide with these previous observations, considering the dominance of *Cetobacterium* and the reduced alpha diversity in larger fish. Other examples of high abundant bacteria (>80% relative abundance) in fish microbiota have been reported previously [18,20].

## 5. Conclusions

Although there are significant differences in the intestinal microbiota of small and large fish of *A. gigas*, members with greater relative abundance have been found to be present in similar proportions. Our results also highlight the dominance of the genus *Cetobacterium* in both sizes examined. The contribution of this genus to host nutrition and health should be explored in future studies. These results can be useful developing a sustainable set of this species.

## Figures and Tables

**Figure 1 animals-08-00189-f001:**
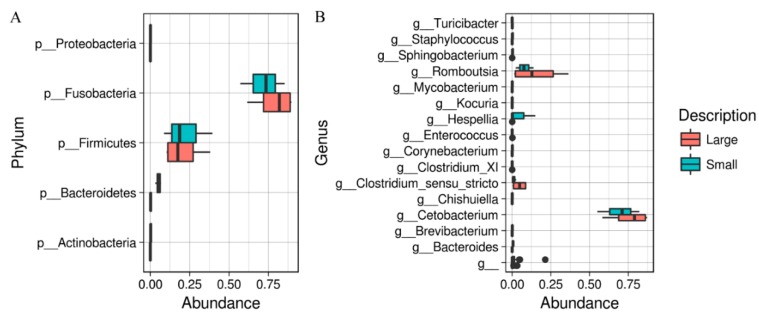
Distribution of bacterial taxa composition in the intestinal microbiota of *Arapaima gigas*; (**A**) phyletic distribution for both size groups; (**B**) genera distribution for both size groups.

**Figure 2 animals-08-00189-f002:**
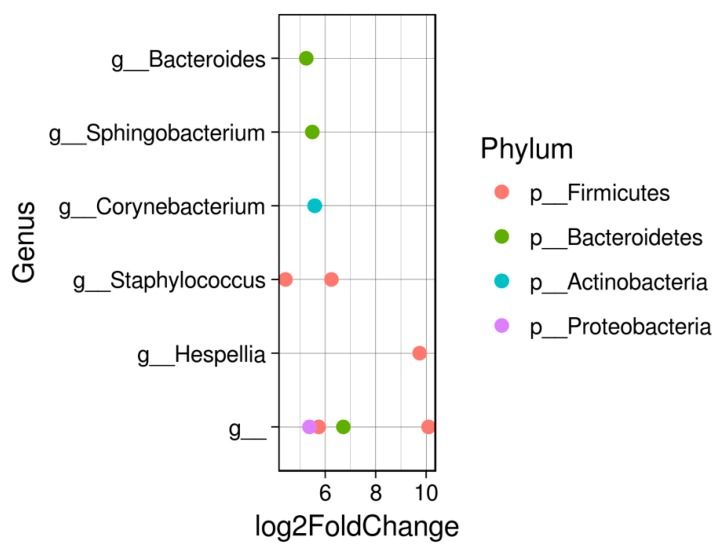
Analysis identifying statistically significant differences in intestinal microbiome components of *Arapaima gigas*. The positive values of the log2-fold change indicate that the operational taxonomic units (OTUs) are significantly higher in small fish than in large fish, considering *p* < 0.05.

**Table 1 animals-08-00189-t001:** Alpha diversity indices of small and large *Arapaima gigas* individuals.

Group Size	Sample	Alpha Diversity Index *			
		OTUs	Chao1	Shannon	Simpson
Small	C1	51	60	2.087	0.823
	C2	50	53	1.915	0.759
	C4	60	63	2.291	0.818
	**Est. mean**	54 ^a^	59 ^a^	2.098 ^a^	0.800 ^a^
Large	7	33	35	1.527	0.638
	8	37	38	1.618	0.659
	12	42	49	1.758	0.748
	18	40	40	1.932	0.793
	**Est. mean**	38 ^b^	40 ^b^	1.709 ^b^	0.709 ^a^

* Different letters superscripts (a or b) indicate significant differences in alpha diversity between groups, considering *p <* 0.05.

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
