# Peer review of "Cetobacterium Is a Major Component of the Microbiome of Giant Amazonian Fish (Arapaima gigas) in Ecuador"

_animals, 2018, doi:10.3390/ani8110189_

Round 1

Reviewer 1 Report

I have made numerous comments on the PDF of this MS, all of which are intended to improve the quality of the report. The major issue that I have is not with the quality of the research, but with the logical jump that the results "highlight ...  Cetobacterium as a produced of Vitamin B-12."  While you have demonstrated that members of this genus are present in the guts, you have not shown that in this habitat is contributes significant amounts of Vitamin B-12.  I urge you to use this paper as a proof of concept: that Cetobacterium is present and then follow up with in vitro research showing that members of this genus (isolated from freshly caught A. gigas) can produce Vitamin B-12.

In spite of my comments I would like to see this paper revised and resubmitted.

Author Response

We would like to thank the reviewer for their detailed comments and suggestions for the manuscript. We believe that the comments have identified important areas which required  improvement.

We have made corrections in Introduction and Conclusions. Corrections are showed in blue.

Introduction:

In this context, understanding the composition of the microbiota could provide relevant information regarding the management of the food requirements of the species in the development of a sustainable aquaculture. It could be also useful for manipulating microbiota during different growth stages in aquaculture systems to prevent pathogenic infection or to improve nutrition. Microbiota modulation has been studied by probiotic supplementation; however, the use of autochthonous probiotics seems to be more beneficial than the use of probiotics from other sources of isolation [4]. The aim of this study was to explore the composition of the bacterial community of Arapaima gigas reared in Ecuador using high-throughput sequencing.

Conclusions:

Although there are significant differences in the intestinal microbiota of small and large fish of A. gigas, members with greater relative abundance are present in similar proportions. Our results also highlight the dominance of the genus Cetobacterium in both sizes examined. The contribution of this genus to host nutrition and health should be explored in future studies. These results can be useful developing a sustainable of this species.

Reviewer 2 Report

Dear Authors

This manuscript describes the specificity of microbial composition of Arapaim gigas intestinal contents and this is estimated to be valuable, as the first report concering this Amazonian animal. By following the previous report, we have already known that several bacterial species have an ability involved in V-12 production in gut of freshwater fishes, and your results today are giving the scientific information that the similar microbiological processes are detected also in Amazonian freshwater fish. This is one of the novel observation. 

Please give attention to folowing several issues.

The order of the tables and figures site shoud be Table 1, Fig. 1 and Fig. 2. And in Table 1, "Observed" should be replaced by "OTUs", shouldn't it?

The small and large fish were used and statistically analyzed, respectively in this study. Are there any special reasons for two sizes? If you have some logical or perspective reasons, I recommend you to note them in Introduction. In the present manuscript, the insight into comparison between fish sizes (Alpha diversity index; Table 1 and microbiome components; Figure 2) seems weak so it is better to discuss more carefully and clearly. 

The detail of the way how to collect the intestinal contents must be essential as the scientific report. Were they collected as the feces from the fish? Or were they collected directly from the intestine after the dissection process? This is a simple question but important, I think. I'd like to recommend you to detail the collecting methods from the microbiological aspect.

Best regards,

Reviewer

Author Response

We would like to thank the reviewer comments and suggestions for the manuscript. 

The order of the tables and figures site shoud be Table 1, Fig. 1 and Fig. 2. And in Table 1, "Observed" should be replaced by "OTUs", shouldn't it?

The order was changed as suggested. OTUs were included instead of Observed in Table 1.

The small and large fish were used and statistically analyzed, respectively in this study. Are there any special reasons for two sizes? If you have some logical or perspective reasons, I recommend you to note them in Introduction. In the present manuscript, the insight into comparison between fish sizes (Alpha diversity index; Table 1 and microbiome components; Figure 2) seems weak so it is better to discuss more carefully and clearly. 

In the revised version, we include new sentences in intro and discussion:

Introduction:

In this context, understanding the composition of the microbiota could provide relevant information regarding the management of the food requirements of the species in the development of a sustainable aquaculture. It could be also useful for manipulating microbiota during different growth stages in aquaculture systems to prevent pathogenic infection or to improve nutrition. Microbiota modulation has been studied by probiotic supplementation; however, the use of autochthonous probiotics seems to be more beneficial than the use of probiotics from other sources of isolation [4]. The aim of this study was to explore the composition of the bacterial community of Arapaima gigas reared in Ecuador using high-throughput sequencing.

Discussion.

Previous studies have described the changes in dominant populations of fish microbiota during growth [18]. In the case of coho salmon, studies suggest that stable gastrointestinal microbiota could be acquired in the juvenile stage, where microbiota showed similar composition in several sizes examined [19]. Our results are coincident with these previous observations, considering the dominance of Cetobacterium and the reduced alpha diversity in larger fish. Other examples of high abundant bacteria (>80% relative abundance) in fish microbiota have been reported previously [18, 20].

The detail of the way how to collect the intestinal contents must be essential as the scientific report. Were they collected as the feces from the fish? Or were they collected directly from the intestine after the dissection process? This is a simple question but important, I think. I'd like to recommend you to detail the collecting methods from the microbiological aspect.

In the revised version, we include the following sentence: Briefly, the digestive tract was aseptically separated from the abdominal cavity with a scalpel, and the hindguts were squeezed to remove and collect the intestinal contents.

Round 2

Reviewer 1 Report

With the changes I suggested the MS is ready for publication.  One of the concepts that I like is that there is a difference in the intestinal flora in young and adult Arapaima giga.  This paper could encourage others to look for differences in additional species and, perhaps more importantly, to examine the shift with age and with diet.

Author Response

We thanks the reviewer work.

The revised version 2 shows the corrections in red letters.